# Low-Velocity Impact Response of Auxetic Seamless Knits Combined with Non-Newtonian Fluids

**DOI:** 10.3390/polym14102065

**Published:** 2022-05-19

**Authors:** Vânia Pais, Pedro Silva, João Bessa, Hernâni Dias, Maria Helena Duarte, Fernando Cunha, Raul Fangueiro

**Affiliations:** 1Fibrenamics, Institute of Innovation on Fiber-Based Materials and Composites, University of Minho, 4800-058 Guimarães, Portugal; pedrosilva@fibrenamics.com (P.S.); joaobessa@fibrenamics.com (J.B.); fernandocunha@det.uminho.pt (F.C.); rfangueiro@dem.uminho.pt (R.F.); 2Centre for Textile Science and Technology (2C2T), University of Minho, 4800-058 Guimarães, Portugal; 3Playvest-Nextil Group Sports Division, Rua das Austrálias, 4705-3226 Braga, Portugal; hernadias@gmail.com (H.D.); maria.duarte@playvest.pt (M.H.D.)

**Keywords:** auxetic, non-Newtonian fluids, low-velocity impact, personal protection

## Abstract

Low-velocity impact can cause serious damage to the person or structure that is hit. The development of barriers that can absorb the energy of the impact and, therefore, protect the other side of the impact is the ideal solution for the pointed situation. Auxetic materials and shear thickening fluids are two types of technologies that have great capabilities to absorb high levels of energy when an impact happens. Accordingly, within this study, the combination of auxetic knits with shear thickening fluids by the pad-dry-cure process was investigated. It was observed that, by applying knits with auxetic patterns produced with denser materials and combined with the shear thickening fluids, high performance in terms of absorbed energy from puncture impact is obtained. The increment rates obtained are higher than 100% when comparing the structures with and without shear thickening fluids.

## 1. Introduction

Any structure that suffers an impact can be damaged, reversible, or irreversible, depending on factors such as the geometry, the mass, and the velocity of the striker. The frequency of the impact will also affect the result. Low-velocity impacts usually occur in the range of 1–10 m/s depending on the properties of the projectile and the defensive structure. In addition, the properties of the structure will influence the impact consequences [1,2]. These structures can act as a barrier to protect an object or a person located on the opposite side of the impact and can be applied in areas such as sports, automotive, or personal protection [2,3,4]. In an ideal situation, this structure should absorb as much energy as possible, so that the user or the object that is being protected feels the smallest impact possible [5].

Auxetic materials present a special behaviour when subjected to mechanical forces. These materials have a negative Poissons’ ratio since, when they expand laterally, an expansion occurs also in the opposite direction. On the other side, when they are compressed laterally, compression on the opposite side also occurs [3]. Auxetic materials present exceptional properties such as increased mechanical properties, shear resistance, indentation resistance, surface fitting ability, and increased breaking tenacity and energy absorption [3,5,6,7]. Due to the mentioned topics, auxetic materials are widely applied in the scope of protection.

Shear thickening fluid (STF) is a non-Newtonian fluid that has low viscosity values when under normal conditions. However, when the shear rate is increased to a critical value, the viscosity also increases at a high rate. At this point, the fluid changes to a solid-state [6,8,9]. Due to this ability, the non-Newtonian fluid can have an important role in impact dynamics [10], since it positively influences the mechanical properties and increases parameters performance such as energy absorption [9].

Due to these special behaviours of auxetic materials and non-Newtonian fluids, the combined application of these two technologies in the protection has much potential. The research presented in this paper aims to investigate the mechanical behaviour of auxetic structures produced in a seamless loom as well as the protective effect of this auxetic structure when combined with non-Newtonian fluids. To develop the auxetic seamless knits, two distinct patterns produced with conditions previously optimized by the group were studied. The amount of non-Newtonian fluid, as well as its application by the pad-dry-cure process, was also studied to obtain a hybrid system with protective properties. Beyond the potential of the application of these two combined technologies, the auxetic structures produced by seamless technology are also an innovative concept and have the high advantage of their reduced weight and high flexibility—when compared to composite structures. Therefore, auxetic knits as well as their combination with STF were optimized within this study in order to obtain materials with increased mechanical performance.

## 2. Materials and Methods

### 2.1. Materials

The auxetic knits were produced on a seamless Santoni loom, model Top 2 by Playvest (Braga, Portugal). The knits are composed of polyamide (PA) and elastane. The PA is produced by air-jet spinning. Two different types of PA were studied—A 78/68×2 and PA 78/68×3 and two different knit stitch tightening were studied for each type of polyamide, with P0 being the least tight and P-15 being the tightest. The two auxetic patterns studied are represented in Figure 1. A total of six different combinations was obtained, as represented in Table 1. The non-Newtonian fluids were acquired at Polyanswer.

### 2.2. Methods

The auxetic behaviour was studied by imposing the extension of 4 cm of the knits in wales and course directions, separately, with the measurement of the extension or contraction between two specific points in the opposite direction.

The weight of the samples was calculated by dividing the mass of the sample mat by its effective area. The friction coefficients were measured using a binary sensor. This equipment was developed in the University of Minho [11] present on the Frictorq equipment. During this assay, a constant rotational motion was applied between the sample under study and a counter-fabric under a uniformly distributed contact force. The air permeability was measured using an air permeability tester from TEXTEST instruments (Zurich, Switzerland, model FX 3300 and a pressure of 200 Pa was applied with the measurement of the airflow that can pass through the structure.

The tensile, tear, and punction tests were performed by using a dynamometer Mesdanlab Twistronic (Hansfield, Salfords, England). On the tensile test, a tension movement was applied under a constant rate and the maximum force and maximum elongation were measured, according to the standard NP EN 13934-2. The tear test was performed according to the standard ASTM D 2261–2017 and the average of the five highest strength values was determined. The puncture test measures the maximum force and energy absorbed during puncture with a piercing element according to the standards ISO 13996: 1999 and EN 863. The burst test follows the standard NP EN ISO 13938-1 and the burst resistance is measured under constant pressure.

The STF were impregnated on the auxetic knits by the pad-dry-cure process. In this process, the knit goes through a bath and then through two rollers, controlled by pressure and speed.

## 3. Results

### 3.1. Auxetic Behaviour

The knits produced on the seamless loom without STF were analyzed in order to see if they have auxetic behaviour. The samples with both patterns in the study produced with PA 78/68×2 with P0 knit stitch were evaluated. Accordingly, the measurement of the extension or contraction in the opposite direction in which the extension is executed was performed. The results are represented in Figure 2. Note that the wales and courses mentioned in the legend refer to the opposite direction in which the movement is imposed. The positive values represent the situation where the extension in one direction is followed by the extension in the opposite direction. Accordingly, the positive values represent the samples that have auxetic behaviour. Concerning the knits with the L pattern, it is observed that when the extension is imposed in the course direction, the opposite direction, i.e., the wale direction, always has a positive value. Thus, the L pattern has auxetic behaviour in the direction. In knits with the P pattern, the wales also have higher values; however, the final value of the extension is equal to zero. Thus it can generally be concluded that the L pattern has a greater prevalence of auxetic behaviour than the P pattern. This auxetics nature only happens when the extension happens in the course direction, being the wales direction the side that expands laterally. Hu et al. also concluded that, when the strain happens in the course direction, a higher auxetic behaviour is observed. The reason pointed out by the authors for this behaviour is the fact that, in the wale direction, the stripes are closer. Accordingly, when the course direction is extended, a higher transversion expansion effect occurs [12].

### 3.2. Structural and Mechanical Characterization

The auxetic knits without STF were characterized and analysed to study their protective behaviour and their comfort parameters. The knits weight, specific air permeability (SAP), and the friction coefficient values (COF) of the L and P patterns produced in the several conditions in this study are shown in Figure 3. Regarding the weight values, it is observed that the L pattern has smaller values. Accordingly, during the production of the L knit, less raw material is required. It is also observed that when the knits are produced with a denser PA-PA 78/68×3-higher weight values are obtained. This result is logical since, in this situation, a higher amount of material is present in the same area. The change in the degree of tightness does not seem to promote changes in weight as significantly as the other parameters since no behavioural trend is observed. These results will directly influence the SAP values since denser structures limit the passage of air with greater impact due to the higher obstruction of pores [13]. The obtained results fall under this statement since the samples with denser PA have smaller SAP values. In COF values it is observed that the L pattern has higher values, regardless of production conditions. When a denser PA is applied it is also observed that the COF values are higher. However, the difference between the several values is small and in some samples is within the range of standard deviation values. Accordingly, it can be concluded that denser PA promotes the production of a more compact knit, which, in turn, will decrease the amount of air able to pass through the structure and will increase the friction intensity. The weight and air permeability parameters will affect the comfort parameters and they will influence properties like moisture management and thermoregulation [14]. Additionally, the friction will also affect the comfort parameters since when the skin contacts a surface, a touch perception occurs. Moreover, higher COF values may cause damage to the skin [15]. The obtained values fall under the ones referred by other authors [15,16].

The mechanical behaviour of the knits was also studied and were applied to different types of tests a tensile test, a tear test and a burst test. For each analysis, the corresponding maximum force achieved was analysed. The obtained results are represented in Figure 4. In the tensile test, regarding the L pattern, the maximum force values achieved are very similar in both knit directions—the wale and the course. On the other hand, in the P pattern differences were observed depending on the direction in which the tension is applied, obtaining higher values in the course direction. In Figure 1 it is possible to see that the L pattern has a very similar structure in both directions. However, the P pattern has different points depending on the observation direction. Due to this difference, the structure will have different behaviours depending on the direction in which the tension is applied. Yet, despite the P pattern having the highest force values in the course direction, it also has the smallest values in the wale direction. Accordingly, the L pattern may have much potential due to its intermediate values, not compromising its effectiveness in the wale direction. Concerning the types of PA, it is observed that the denser PA has the highest values of maximum force. Thus this compact arrangement will positively reinforce the structure, allowing it to achieve better mechanical performance. It is also important to note that the knit with the P pattern has a maximum force value in the course direction that is much higher than the other samples. This difference must be related to production problems instead of some standout mechanical properties of this sample, since this result is not observed in the other analysis and has no logical reason to happen. Accordingly, this result should be ignored in the analysis of this study. In the tear test, these differences imposed by the different directions were less noted; however, the same conclusions were observed, i.e., higher maximum forces are achieved when the PA 78/68×3 is applied. In the burst test, once again, the denser sample has the higher force values. Besides that, it was observed that the L pattern has much higher force values. In this final test, both knit directions are evaluated at the same time when executing one test since the assay is performed using a round sample. Accordingly, the differences previously observed in both directions of the P pattern will be conjugated in the final result. Due to this, the burst results with knits with the P pattern had much smaller force values than knits with the L pattern. Generally, concerning the mechanical parameters of the knits, it can be concluded that denser materials positively impact the achievement of higher force values. Samples with a tighter stitch seem to generally have higher force values, however, with a low impact.

### 3.3. Combination of Auxetic Knits with STF

The auxetic knits were combined with the STF to increase their protection capability. Other authors also studied the combination of these fluids with other materials to boost their protective properties. Santos et al. proved that the addition of STF to Kevlar fabrics improves the performance of these substrates under impact tests [17]. Lu et al. and Santos et al. also demonstrated that the combination of STF with materials like wale-knitted spacer fabrics and Kevlar woven samples, respectively, can be applied as a material for personal protection [18,19]. Within the present study, STF were combined with the auxetic knits by the pad-dry-cure process. The impregnation rates changed from 60 to 70%. The knits with L and P patterns, before and after STF impregnation, are shown in Figure 5.

Once more, tensile tests were performed to see the mechanical behaviour of the samples. However, in this turn, the absorbed energy was also calculated to see the potential of the obtained materials to be applied in protective applications. The materials’ ability to absorb energy is important in protection products due to the increase in the distribution of loads throughout the hierarchical architecture [20]. The graphical results obtained by knits with the L and P patterns are represented in Figure 6 and Figure 7, respectively. When performing the tensile tests, it was observed that the knits impregnated with STF could stretch more. This higher stretch capability boosts the achievement of higher absorbed energy values. It was observed that when knits were combined with STF, higher maximum force and absorbed energy values were obtained. The only exception was the P pattern in the course direction. However, the incongruity of this sample was already mentioned in the analysis of the mechanical behaviour of the knits without STF. When comparing the L and P pattern, it is observed that the P pattern has higher energy absorbed values, making the knits stand out where denser PA was applied. This result follows the ones mentioned before, where it was concluded that applying higher amounts of mass in the same area unit increases the capability of the samples to support higher forces in the tensile movement. Moreover, as a consequence of this property, the samples with denser material will have the ability to absorb higher amounts of energy and much more potential to be applied as a protective material/product. Yan et al. also studied the influence of material density on protective applications’ performance. In the mentioned study it was observed that when materials have higher densities they can reach higher forces when burst and puncture tests are performed. The higher amount of fibers and bonding points per unit of the area of materials with higher densities are the reasons given in the study for achieving higher mechanical behaviours [21].

To simulate a real low-velocity impact, puncture tests were performed with the calculation of maximum force and absorbed energy. The obtained results of knits with the L and P patterns—with and without STF—are represented in Figure 8. Once more it was observed that the addition of STF will increase the mechanical and protective behaviour of the samples in this study and that the P patterns have a higher performance in terms of mechanical behaviour. The samples with denser PA and impregnated with STF generally had higher values of energy absorbed, especially in the situation in which the stitch is tighter. These two combinations are the perfect match to correspond to the points mentioned by Yan et al., i.e., higher amount of fibers and bonding points per unit of area [21]. This combination will increase the protective effect of the samples.

## 4. Conclusions

In this research, a combination of auxetic knits with STF was studied. First, the auxetic knits were characterized, and it was concluded that denser materials increase the mechanical performance of the final structure. When both technologies are combined through the pad-dry-cure process, a higher increase in mechanical behaviour happens. Moreover, puncture tests were performed to simulate the low-velocity impact and it was concluded that the absorbed energy increases with the presence of STF, being more visible in the knit with the P pattern. The increase of materials density, as well as the tightness stitch, will increase the protective behaviour mentioned. The product resulting from the combination of the technologies in this study is very light when compared to the most protective products on the market. Accordingly, this is a promising approach to be applied in personal protection applications.

## Figures and Tables

**Figure 1 polymers-14-02065-f001:**
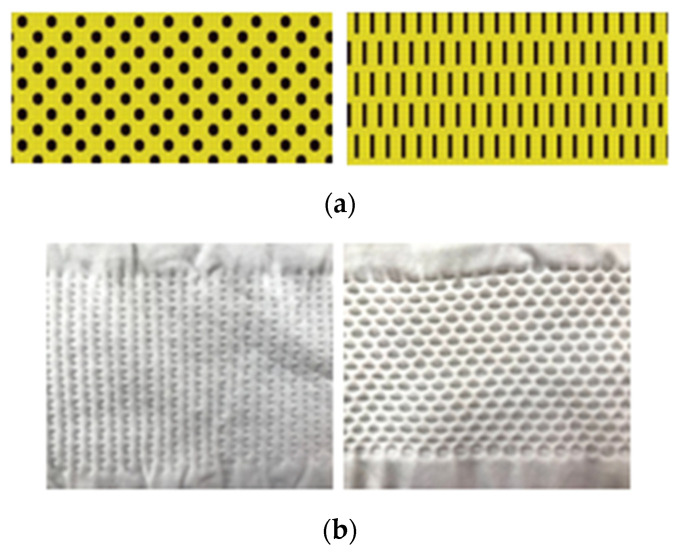
Technical drawing of pattern L and P (**a**) and pattern L and P on seamless knit (**b**).

**Figure 2 polymers-14-02065-f002:**
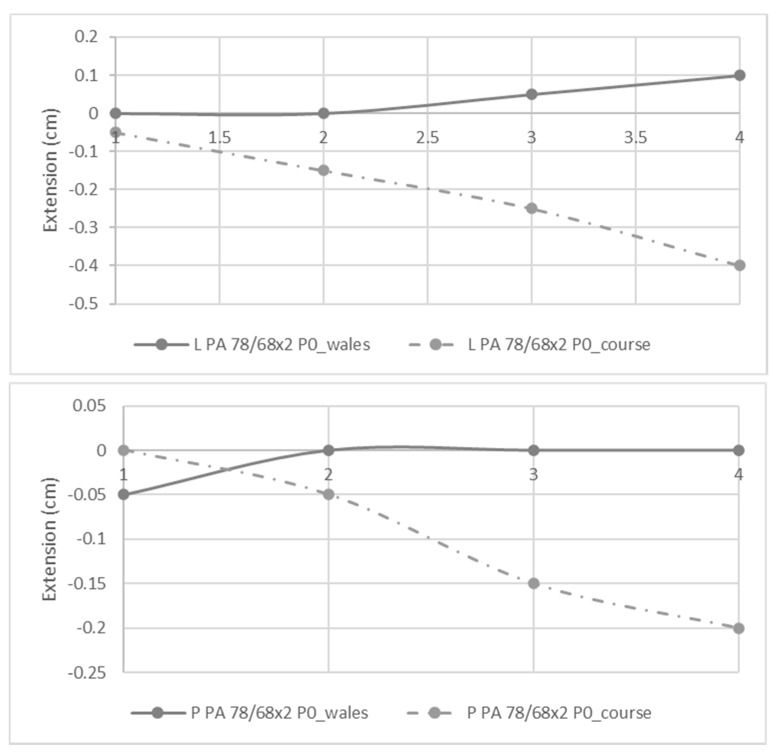
Extension of knits with L and P patterns in wales and course direction when imposed extension in the opposite direction. The *x*-axis represents the strain being applied.

**Figure 3 polymers-14-02065-f003:**
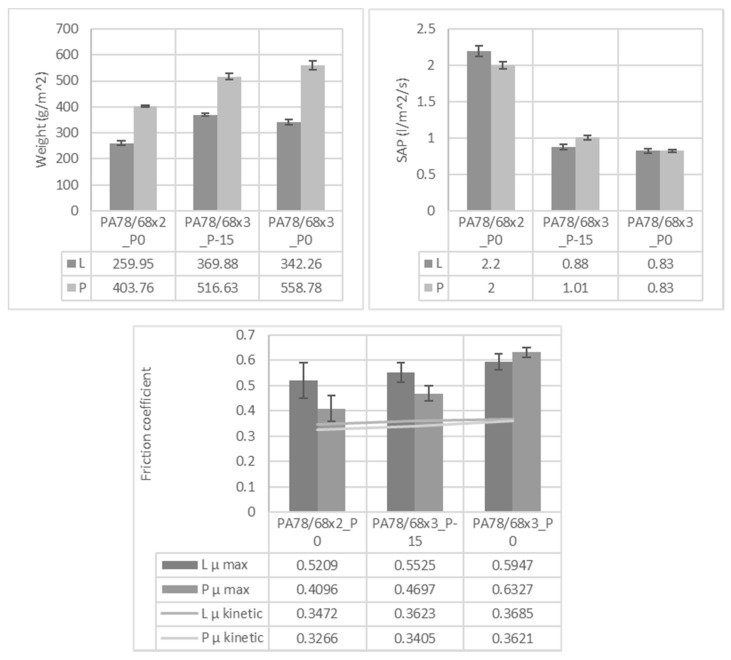
Weight, SAP and friction coefficient of knit samples.

**Figure 4 polymers-14-02065-f004:**
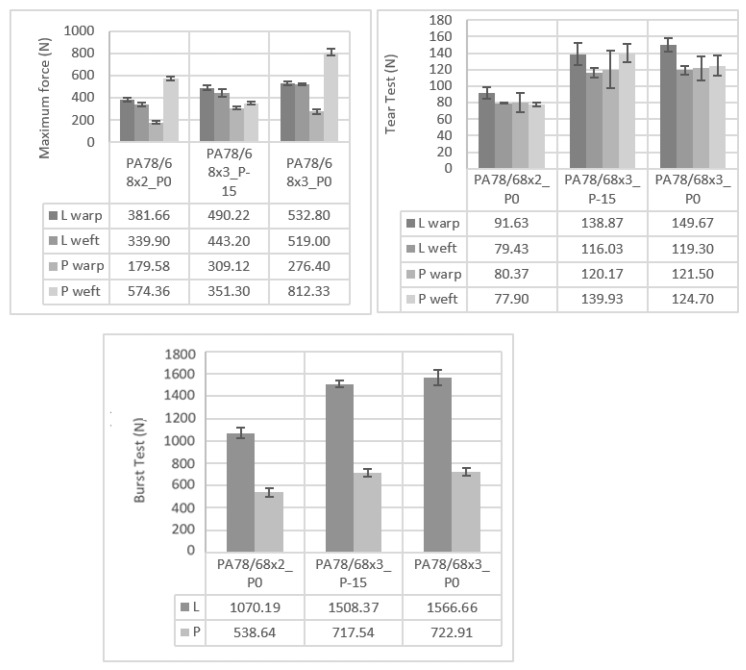
Maximum forces obtained in the tensile test, tear test and burst test.

**Figure 5 polymers-14-02065-f005:**
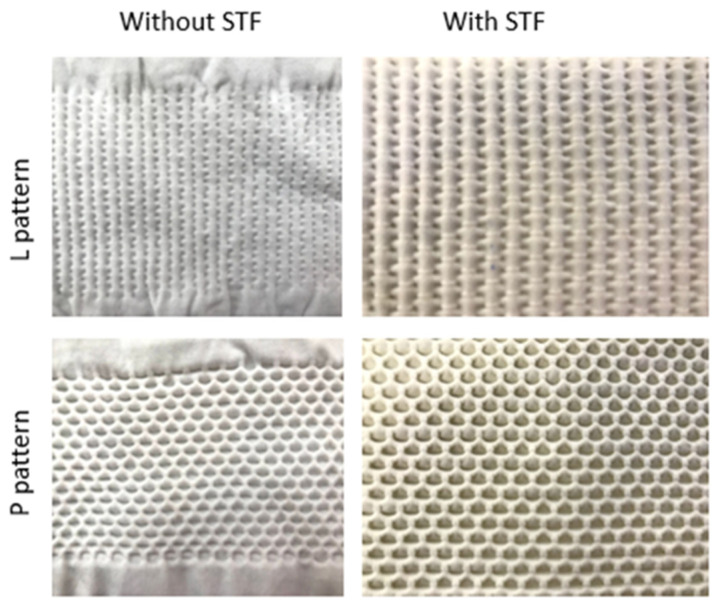
Knits with L and P patterns, with and without STF.

**Figure 6 polymers-14-02065-f006:**
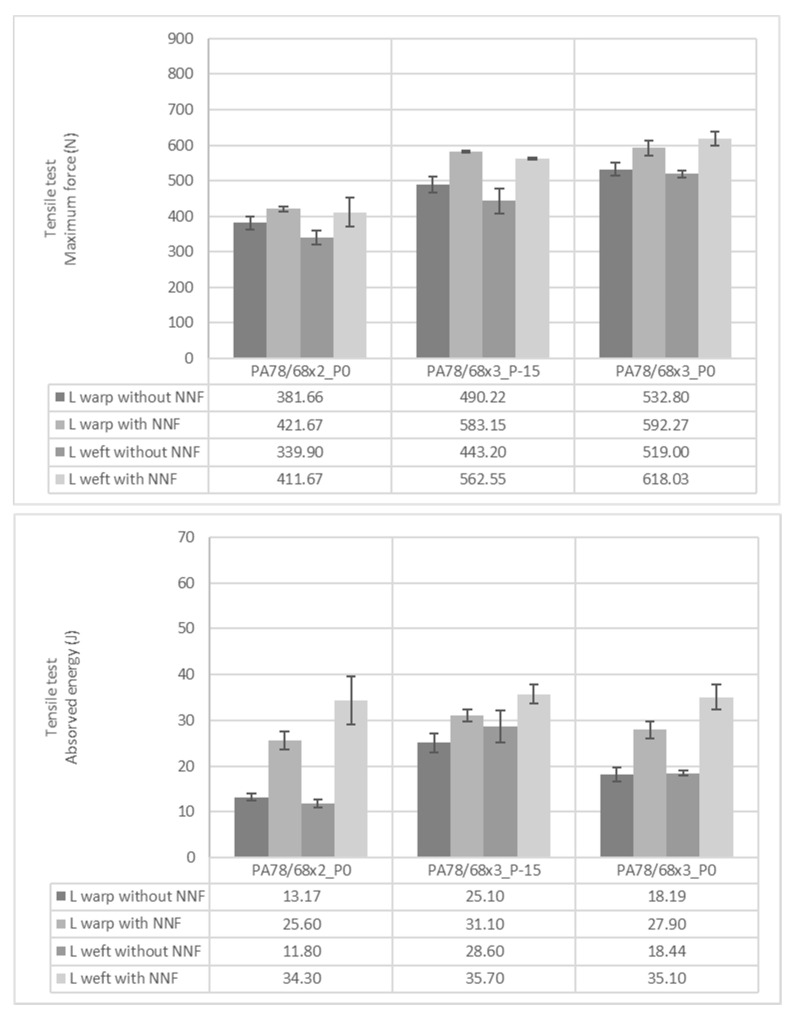
Maximum force and absorbed energy of knits with L pattern obtained on tensile tests.

**Figure 7 polymers-14-02065-f007:**
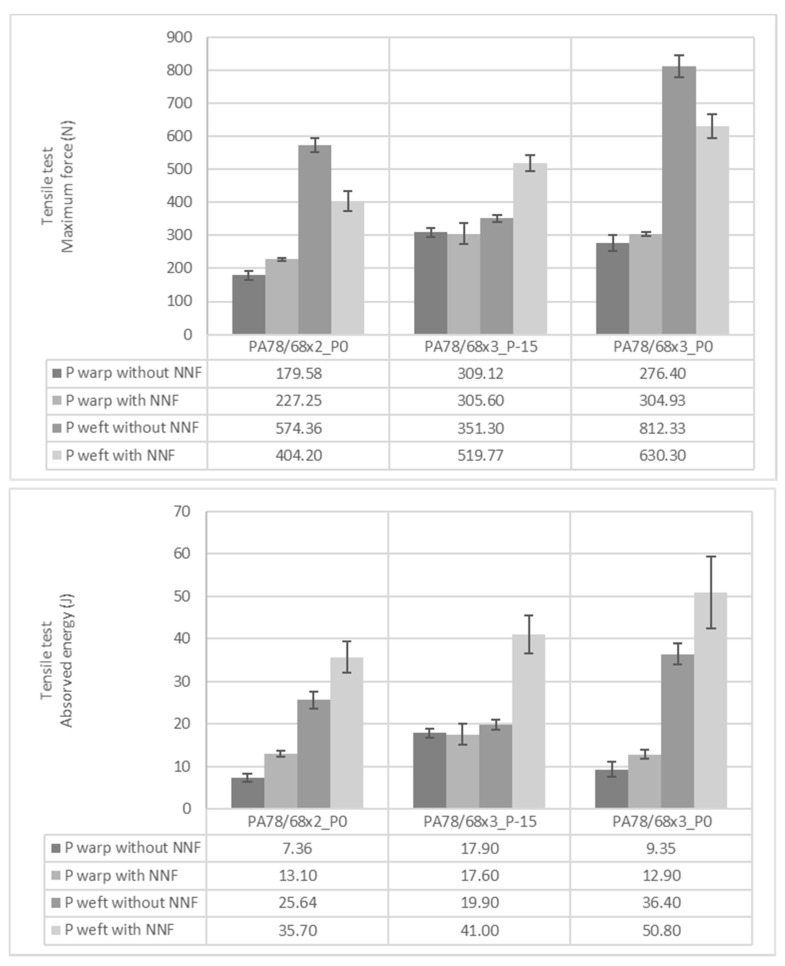
Maximum force and absorbed energy of knits with P pattern obtained on tensile tests.

**Figure 8 polymers-14-02065-f008:**
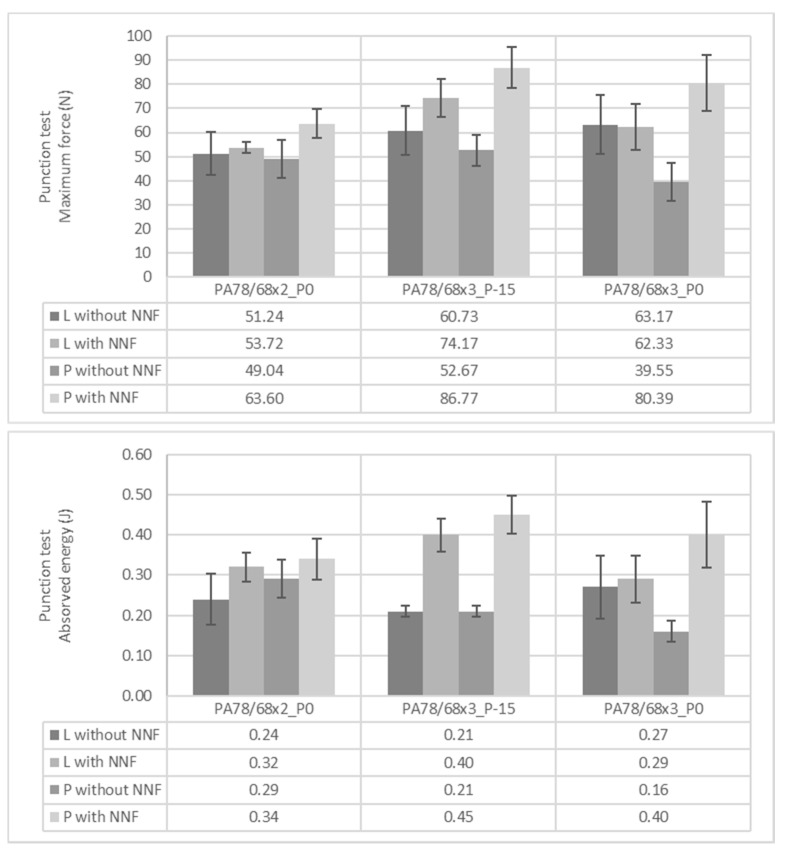
Maximum force and absorbed energy of knits with L and P pattern obtained on punction tests.

**Table 1 polymers-14-02065-t001:** Knits studied in the present study.

Pattern	Polyamide	Knit Stitch	Reference
L	PA 78/68×2×2	P0	L_PA 78/68×2_P0
	PA 78/68×3	P-15	L_PA 78/68×3_P-15
		P0	L_PA 78/68×3_P0
P	PA 78/68×2×2	P0	P_PA 78/68×2_P0
	PA 78/68×3	P-15	P_PA 78/68×3_P-15
		P0	P_PA 78/68×2×2_P0

## Data Availability

Not applicable.

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
