# Peer review of "Low-Velocity Impact Response of Auxetic Seamless Knits Combined with Non-Newtonian Fluids"

_polymers, 2022, doi:10.3390/polym14102065_

Round 1

Reviewer 1 Report

Although auxetic materials and shear thickening fluids (STF) are not new, and the use of each of them for impact protection has been earlier suggested, the novelty of this submitted work is in combining both materials. Specifically, the auxetic textiles are being used in a way analogous to reinforcing material while the STF being analogous to the surrounding matrix material, with the aim of producing a mechanically robust protective clothing against impact. To do so, there must be an additional layer of material (hopefully very thin but sufficiently flexible and strong) that can enclose textile-SFT system in order to prevent the STF from leaking out. Can the authors suggest the following:

(a) what enclosing material that is sufficiently thin (since the auxetic textile and the presence of SFT would introduce certain thickness). but sufficiently strong to prevent leakage of the SFT and at the same time flexible enough to permit the workers, who wear this protective clothing, to do physical work.

(b) since there is a need to have an additional layer to prevent leakage of STF, how to create air flow underneath the clothing for worker comfort?

(c) How to ensure that fast body motion or contact with vibrating machinery does not cause too much increase in viscosity in the STF, so that the workers' efficiency is not slowed down.

Due to the rapid increase of auxetic research it is understandable that no literature review in auxetics can be fully incorporated. As such, the authors are advised to refer the readers to the only two available monographs in auxetics  https://doi.org/10.1007/978-981-287-275-3    and  https://doi.org/10.1016/C2016-0-04399-1 , which contain enormous references in auxetic systems for readers to refer.

Author Response

Dear reviewer,

Thank you very much for your kind suggestions. 
I'm sending in an attachment with the response to each comment.

Best regards,
Vânia Pais

Reviewer 2 Report

The following major remarks appear:

1 - some important papers in the subject are missing: Int. J. Solids Struct. 2006, 43, 1746–1763;
Compos. Struct. 2012, 94, 2326–2336;Energies 2021, 14(1), 214; https://doi.org/10.3390/en14010214;Study. Mater. Des. 2010, 31, 1216–1230;Compos. Part B: Eng. 2013, 47, 267–277

2 - all figures are of very low quality

3 - Section 1 - please state precisely the novelty and structure of the paper

4 - commercialism like 'Playvest S.A.', 'Kevlar®' must be cancelled from scientific paper

Author Response

(The authors gave the same response as above.)

Round 2

Reviewer 2 Report

Thank you.